# Effect of *Lactobacillus rhamnosus* hsryfm 1301 Fermented Milk on Lipid Metabolism Disorders in High-Fat-Diet Rats

**DOI:** 10.3390/nu14224850

**Published:** 2022-11-16

**Authors:** Hengxian Qu, Lina Zong, Jian Sang, Yunchao Wa, Dawei Chen, Yujun Huang, Xia Chen, Ruixia Gu

**Affiliations:** 1College of Food Science and Technology, Yangzhou University, Yangzhou 225000, China; 2Jiangsu Key Laboratory of Dairy Biotechnology and Safety Control, Yangzhou University, Yangzhou 225000, China; 3Realab Biotechnology Co., Ltd., Beijing 100000, China

**Keywords:** probiotics, lipid metabolism, transcriptomics, metabolomics, bioinformatic

## Abstract

To further explore and improve the mechanism of probiotics to alleviate the disorder of lipid metabolism, transcriptomic and metabolomic with bioinformatic analysis were combined. In the present study, we successfully established a rat model of lipid metabolism disorder using a high-fat diet. Intervention with *Lactobacillus rhamnosus* hsryfm 1301 fermented milk resulted in a significant reduction in body weight, serum free fatty acid and blood lipid levels (*p* < 0.05), which predicted that the lipid metabolism disorder was alleviated in rats. Metabolomics and transcriptomics identified a total of 33 significantly different metabolites and 183 significantly different genes screened in the intervention group compared to the model group. Comparative analysis of Kyoto Encyclopedia of Genes and Genomes (KEGG) pathway annotations identified a total of 61 pathways in which differential metabolites and genes were jointly involved, with linoleic acid metabolism, glycine, serine and threonine metabolism and glutamatergic synapse in both transcriptome and metabolome being found to be significantly altered (*p* < 0.05). *Lactobacillus rhamnosus* hsryfm 1301 fermented milk was able to directly regulate lipid metabolism disorders by regulating the metabolic pathways of linoleic acid metabolism, glycerophospholipid metabolism, fatty acid biosynthesis, alpha-linolenic acid metabolism, fatty acid degradation, glycerolipid metabolism and arachidonic acid metabolism. In addition, we found that *Lactobacillus rhamnosus* hsryfm 1301 fermented milk indirectly regulates lipid metabolism through regulating amino acid metabolism, the nervous system, the endocrine system and other pathways. *Lactobacillus rhamnosus* hsryfm 1301 fermented milk could alleviate the disorders of lipid metabolism caused by high-fat diet through multi-target synergy.

## 1. Introduction

High-fat diet is the main cause of lipid metabolism disorders. High intake of lipids exceeds the body’s metabolic capacity, which contributes to the rising incidence of obesity, hyperlipidemia, fatty liver and type II diabetes [1]. Currently, controlling dietary intake and forming good lifestyle habits are the most important modalities in treatment of lipid metabolism disorders [2]. There are also some drugs that can be used for treatment [3,4]. However, it is difficult to achieve a better therapeutic effect with a single treatment method. The liver is the main site of lipid metabolism in the body, and it is also a key organ in lipid metabolism research. With the introduction of the “intestine–liver axis” theory and a large number of studies, probiotic intervention has become a new direction for adjunctive treatment of lipid metabolism disorders [5].

Probiotics can alleviate lipid metabolism disorders by inhibiting overgrowth of harmful intestinal bacteria, maintaining intestinal microecological balance, absorbing intestinal cholesterol and regulating lipid metabolism [6]. Probiotics were also able to directly or indirectly regulate the transcript levels of genes related to lipid metabolism in the liver, thus restoring lipid metabolism to homeostasis. The mixed sample of *Lactobacillus delbrueckii* and *Lactobacillus plantarum* was able to affect transcription of liver ACAT and CYP7α1 genes to regulate lipid anabolic processes and reduce lipid levels [7]. *Lactobacillus johnsonii* was able to decrease transcription of stearoyl coenzyme A desaturase-1 and lipoprotein esterase, downregulate transcription of SREBP-1c and fatty acid synthase in the liver to reduce lipid synthesis and enhance transcription of PPARα and acetyl coenzyme A oxidase to enhance lipid degradation, thereby reducing TG, LDL and increasing HDL [8]. With the help of transcriptomics techniques, researchers found that *Lactobacillus casei* was able to affect transcription of more than 700 genes in the liver, upregulating lipid-metabolism-related genes, such as *Acsl1*, *Acaa2*, *Acads*, *gcdH* and *Hadh* [9]. The existing studies are mainly focused on changes in certain specific genes in lipid-metabolism-related pathways or are single transcriptomic studies. However, the process of lipid metabolism is complex. In addition to lipid metabolism, it is also regulated by several pathways, including the endocrine system and the nervous system. The mechanisms by which probiotics regulate lipid metabolism disorders are still not fully understood and require further exploration and improvement.

A previous study found that *Lactobacillus rhamnosus* hsryfm 1301 fermented milk has intestinal flora regulation and hypolipidemic effects. It not only suppressed the abundance of bacteria associated with lipid metabolism disorders in the intestine but also increased the abundance of probiotics, such as *Bifidobacterium* and *Lactobacillus* [10]. Probiotics, including *Lactobacillus rhamnosus* hsryfm 1301 fermented milk, exert an important influence on regulation of lipid metabolism disorders. In this study, we combined metabolomics and transcriptomics to integrate serum metabolites and liver genes in high-fat-diet rats after *Lactobacillus rhamnosus* hsryfm 1301 fermented milk intervention. It is also combined with bioinformatic analysis, such as metabolic pathway enrichment, to explore the interaction between serum metabolites and liver genes, which will further resolve the molecular mechanism of *Lactobacillus rhamnosus* hsryfm 1301 in regulating lipid metabolism disorder. It will also further detail the mechanism of action of probiotics in regulating lipid metabolism.

## 2. Materials and Methods

### 2.1. Probiotic Bacteria and Fermented Milk Preparation

*Lactobacillus rhamnosus* hsryfm 1301 were provided by Jiangsu Key Lab of Dairy Biological Technology and Safety Control, China. After two generations of activation, the strain was inoculated into a skim milk medium at 3% inoculum, incubated at 37 °C for 18 h, then inoculated into whole milk at 3% inoculum and fermented at 37 °C for 72 h to prepare *Lactobacillus rhamnosus* hsryfm 1301 fermented milk.

### 2.2. Animals

Twenty-seven healthy male Wistar rats were purchased from Comparative Medical Center of Yangzhou University, Jiangsu, China. The rats were 6 weeks old and weighing about 218 g at the start of the experiment. All animals were housed under a 12 h light/12 h dark cycle in a controlled room with a temperature of 23 ± 3 °C and a humidity of 50 ± 10%. The animals were acclimated to their new circumstances for one week. Then, rats were randomly divided into 3 groups (n = 9): normal group (C), model group (M) and *L. rhamnosus* hsryfm 1301 treatment group (P). Normal group fed a low-fat diet (LFD: flour 20%, rice flour 10%, corn 20%, drum skin 26%, soy material 20%, fish meal 2%, bone meal 2%), and model group and *L. rhamnosus* hsryfm 1301 treatment group fed a high-fat diet (HFD: 10% lard, 10% egg powder, 1% cholesterol and 0.2% bile salts and 78.8% LFD) for 8 weeks. All rats were allowed free access to food and water. All rats received the following treatments by lavage: normal group and model group rats: milk (1 mL/100 g); *L. rhamnosus* hsryfm 1301 treatment group: *L. rhamnosus* hsryfm 1301 fermented milk (1 mL/100 g, 10^9^ CFU/mL). The body weights of rats were recorded weekly. After 8 weeks, rats underwent 12 h of fasting prior to being anaesthetized and dissected. All rats were euthanized at the anestrus period following anesthesia under 1% sodium entobarbital. Serum was collected by centrifugation at 3000× *g* for 5 min and stored at −20 °C. Livers were removed and stored at −8 °C for subsequent analyses.

### 2.3. Blood Lipid Levels and FFAs in Serum

Total cholesterol (TC) and triglyceride (TG) of supernatant were detected according to standard procedures with the assay kits (Maccura Biotechnology Co., Ltd., Sichuan, China) by use of fully automatic biochemical analyzer (Hitachi, Japan). Contents of free fatty acid (FFA) were determined by the standard manuals supplied with the assay kits (Beijing Solarbio Science & Technology Co., Ltd., Beijing, China).

### 2.4. Metabolomics

#### 2.4.1. Sample Preparation

The samples were thawed at 4 °C and 100 μL aliquots were mixed with 400 μL of cold methanol/acetonitrile (1:1, *v/v*) to remove the protein. The mixture was centrifuged for 20 min (14,000× *g*, 4 °C). The supernatant was dried in a vacuum centrifuge. For LC–MS analysis, the samples were re-dissolved in 100 μL acetonitrile/water (1:1, *v/v*) solvent.

#### 2.4.2. Mass Spectrometry

Analyses were performed using a UHPLC (1290 Infinity LC, Agilent Technologies, Santa Clara, CA, USA) coupled to a quadrupole time-of-flight (AB Sciex TripleTOF 6600).

#### 2.4.3. HPLC

Samples were analyzed using a 2.1 × 100 mm ACQUIY UPLC BEH 1.7 µm column (Waters, Ireland). In both ESI positive and negative modes, the mobile phase contained A = 25 mM ammonium acetate and 25 mM ammonium hydroxide in water and B = acetonitrile. The gradient was 85% B for 1 min and was linearly reduced to 65% in 11 min, and then was reduced to 40% in 0.1 min and kept for 4 min and then increased to 85% in 0.1 min, with a 5 min re-equilibration period employed.

#### 2.4.4. MS/MS Analysis

The ESI source conditions were set as follows: Ion Source Gas1 (Gas1) as 60, Ion Source Gas2 (Gas2) as 60, curtain gas (CUR) as 30, source temperature: 600 °C, IonSpray Voltage Floating (ISVF) ± 5500 V. In MS only acquisition, the instrument was set to acquire over the m/z range 60–1000 Da, and the accumulation time for TOF MS scan was set at 0.20 s/spectra. In auto MS/MS acquisition, the instrument was set to acquire over the *m/z* range 25–1000 Da, and the accumulation time for product ion scan was set at 0.05 s/spectra. The product ion scan is acquired using information-dependent acquisition (IDA) with high sensitivity mode selected. The parameters were set as follows: the collision energy (CE) was fixed at 35 V with ± 15 eV; declustering potential (DP), 60 V (+) and −60 V (−); exclude isotopes within 4 Da; candidate ions to monitor per cycle: 10.

### 2.5. Transcriptomic

#### 2.5.1. RNA Quantification and Qualification

RNA degradation and contamination were monitored on 1% agarose gels. RNA purity was checked using the NanoPhotometer^®^ spectrophotometer (IMPLEN, Westlake Village, CA, USA). RNA concentration was measured using Qubit^®^ RNA Assay Kit in Qubit^®^2.0 Flurometer (Life Technologies, Carlsbad, CA, USA). RNA integrity was assessed using the RNA Nano 6000 Assay Kit of the Bioanalyzer 2100 system (Agilent Technologies, Santa Clara, CA, USA).

#### 2.5.2. Library Preparation for Transcriptome Sequencing

A total amount of 3 μg RNA per sample was used as input material for the RNA sample preparations. Sequencing libraries were generated using NEBNext^®^ Ultra^TM^ RNA Library Prep Kit for Illumina^®^ (Ipswich, MA, USA) following manufacturer’s recommendations, and index codes were added to attribute sequences to each sample. Briefly, mRNA was purified from total RNA using poly-T oligo-attached magnetic beads. Fragmentation was carried out using divalent cations under elevated temperature in NEBNext First Strand Synthesis Reaction Buffer(5X). First strand cDNA was synthesized using random hexamer primer and M-MuLV Reverse Transcriptase (RNase H-). Second strand cDNA synthesis was subsequently performed using DNA Polymerase I and RNase H. Remaining overhangs were converted into blunt ends via exonuclease/polymerase activities. After adenylation of 3′ ends of DNA fragments, NEBNext Adaptor with hairpin loop structure was ligated to prepare for hybridization. In order to select cDNA fragments of preferentially 250~300 bp in length, the library fragments were purified with AMPure XP system (Beckman Coulter, Beverly, NJ, USA). Then, 3 μl USER Enzyme (NEB, Ipswich, MA, USA) was used with size-selected, adaptor-ligated cDNA at 37 °C for 15 min followed by 5 min at 95 °C before PCR. Then, PCR was performed with Phusion High-Fidelity DNA polymerase, Universal PCR primers and Index (X) Primer. At last, PCR products were purified (AMPure XP system) and library quality was assessed on the Agilent Bioanalyzer 2100 system.

#### 2.5.3. Clustering and Sequencing

Clustering of the index-coded samples was performed on a cBot Cluster Generation System using TruSeq PE Cluster Kit v3-cBot-HS (Illumia) according to the manufacturer’s instructions. After cluster generation, the library preparations were sequenced on an Illumina platform and 125 bp/150 bp paired-end reads were generated.

### 2.6. Statistical Analysis

#### 2.6.1. Biochemical Data Analysis

Biochemical data analysis was performed using GraphPad Prism 9 (San Diego, CA, USA). The results are presented as mean ± SD, and the differences between the different samples were analyzed using one-way analysis of variance (ANOVA, Ankara, Tukey).

#### 2.6.2. Differential Metabolite Analysis

The raw MS data (wiff.scan files) were converted to MzXML files using ProteoWizard MSConvert before importing into freely available XCMS software. For peak picking, the following parameters were used: centWave *m/z* = 25 ppm, peakwidth = c (10, 60), prefilter = c (10, 100). For peak grouping, bw = 5, mzwid = 0.025, minfrac = 0.5 were used. In the extracted ion features, only the variables having more than 50% of the nonzero measurement values in at least one group were kept. Compound identification of metabolites by MS/MS spectra with an in-house database established with available authentic standards.

After normalized to total peak intensity, the processed data were uploaded into before importing into SIMCA-P (version 14.1, Umetrics, Umea, Sweden), where it was subjected to multivariate data analysis—orthogonal partial least-squares discriminant analysis (OPLS-DA). Further, 7-fold cross-validation and response permutation testing were used to evaluate the robustness of the model. The variable importance in the projection (VIP) value of each variable in the OPLS-DA model was calculated to indicate its contribution to the classification. Metabolites with the VIP value >1 were further applied to Student’s *t*-test at univariate level to measure the significance of each metabolite; *p* values less than 0.05 were considered as statistically significant.

#### 2.6.3. Differential Expression Analysis and GO Enrichment Analysis

Prior to differential gene expression analysis, for each sequenced library, the read counts were adjusted by edgeR program package through one scaling normalized factor. Differential expression analysis of two conditions was performed using the edgeR R package (3.18.1). The *p* values were adjusted using the Benjamini and Hochberg method [11]. Corrected *p* value of 0.05 and absolute foldchange of 2 were set as the threshold for significantly differential expression.

Gene ontology (GO) enrichment analysis of differentially expressed genes was implemented by the clusterProfiler R package, in which gene length bias was corrected. GO terms with corrected *p* value less than 0.05 were considered significantly enriched by differentially expressed genes.

#### 2.6.4. Bioinformatic Joint Analysis

Principle component analysis (PCA) was performed with SIMCA Version 14.1 using quantitative data of the two omics.

Differentially abundant genes and metabolites were log2 scaled and concatenated into one matrix. Then, correlation coefficients among all the molecules in the matrix were calculated with Pearson algorithm in R Version 3.5.1.

All differentially expressed genes and metabolites were queried and mapped to pathways based on the online Kyoto Encyclopedia of Genes and Genomes (KEGG, http://www.kegg.jp/, accessed on 11 March 2022). Enrichment analysis was also performed. R Version 3.5.1 was used to combine the KEGG annotation and enrichment result of the two omics. Venn diagram and bar plot were drawn.

## 3. Results

### 3.1. Lipid Metabolism Disorders

A high-fat diet caused the symptoms of obesity. The body weight of rats in group M was significantly increased compared with group C (*p* < 0.001) (Figure 1a). The serum levels of FFA, TC and TG were significantly increased (*p* < 0.001) in the group M rats compared with group C (Figure 1b–d), and the high-fat diet caused disorders of lipid metabolism in rats. After the intervention of *Lactobacillus rhamnosus* hsryfm 1301 fermented milk, there was a significant reduction in body weight and serum levels of FFA, TC and TG in rats compared with group M (*p* < 0.05) (Figure 1); especially, the body weight of rats was not significantly different from group C (Figure 1a). This tentatively suggests that *Lactobacillus rhamnosus* hsryfm 1301 fermented milk has a mitigating effect on lipid metabolism disorders caused by high-fat diet.

### 3.2. Metabolomics

A total of 50 differential metabolites were screened in positive and negative ion mode in group P and group M, of which 33 differential metabolites were significantly changed (*p* < 0.05; Figure 2a,b and Appendix A). Compared to group M, 13 differential metabolites, including 1-Stearoyl-2-arachidonoyl-sn-glycerol, L-Glutamine and L-Serine, were significantly increased in group P (*p* < 0.05). Twenty differential metabolites, such as PC(16:0/16:0), all cis-(6,9,12)-linolenic acid and linoleic acid, were significantly decreased (*p* < 0.05).

These differential metabolites were enriched to a total of 104 KEGG pathways, and a total of 30 KEGG pathways were significantly altered (Figure 2c and Appendix A), which included ABC transporters, biosynthesis of unsaturated fatty acids, linoleic acid metabolism, insulin resistance and other pathways related to lipid metabolism. The differential metabolites that were enriched to these pathways were all cis-(6,9,12)-linolenic acid, 1-Stearoyl-2-arachidonoyl-sn-glycerol, etc. In addition, the T cell receptor signaling pathway, B cell receptor signaling pathway, Chemokine signaling pathway, etc., involved in immune system, as well as the NF-kappa B signaling pathway, Rap1 signaling pathway, MAPK signaling pathway, etc., involved in signal transduction, were significantly altered. The differential metabolites that were enriched were mainly 1-Stearoyl-2-arachidonoyl-sn-glycerol.

### 3.3. Transcriptomics

*p* values less than 0.05 and log2 fold change absolute values greater than 1 were used as the criteria for differential significance. A total of 183 genes were significantly changed in group P compared with group M, with 95 upregulated and 88 downregulated (Figure 3a and Appendix A). The cluster analysis showed that the differential genes obtained from the screening could effectively separate group P from group M (Figure 3b), which indicated the rationality of screening differentially expressed genes.

GO functional annotation analysis showed that biological processes, including neutrophil aggregation, complement activation, humoral immune response, regulation of peptidase activity, protein activation cascade and L-serine biosynthetic process, changed significantly. Molecular functions, such as Toll-like receptor 4 binding, have changed dramatically. Cellular components, such as extracellular space, extracellular region part and extracellular region, changed significantly (Figure 3c). KEGG pathway analysis found that 155 KEGG pathways, such as linoleic acid metabolism, glycine, serine and threonine metabolism, etc., were significantly changed (Figure 3d and Appendix A).

### 3.4. Correlation

Comparative PCA analysis showed that the overall expression trends of the samples were the same between transcriptomics and metabolomics groups (Figure 4a,b). Spearman correlation hierarchical cluster analysis of significantly different genes and metabolites showed that they could be distinguished by clustering, while genes and metabolites with similar expression patterns were clustered. This all indicates that transcriptomics and metabolomics are well correlated (Figure 4c).

### 3.5. KEGG Pathway Joint Analysis

There are 61 pathways in which differentially expressed genes in transcriptomics and differentially expressed metabolites in metabolomics are jointly involved (Figure 5a and Table 1), mainly in lipid metabolism, amino acid metabolism, immune system and nervous system pathways. Glycine, serine and threonine metabolism were enriched to the most differential genes and metabolites (Figure 5b). At the same time, three co-enriched pathways, glycine, serine and threonine metabolism, glutamatergic synapse and linoleic acid metabolism, were significantly changed (Figure 5c).

## 4. Discussion

We applied serum metabolomics and liver transcriptome to study the effects of probiotics on lipid metabolism disorders in high-fat-diet rats. Important liver differential genes and serum differential metabolites were identified by combined analysis, as well as the KEGG pathway in which they are jointly involved. The effect of probiotics on lipid metabolism and its related pathways in high-fat-diet rats was more comprehensively explored.

The pathways involved in lipid metabolism, such as linoleic acid metabolism, etc., were first identified by comparative analysis of the KEGG pathway. The linoleic acid metabolism pathway was significantly changed in both transcription and metabolism, with transcription of *Pla2g4c* being significantly downregulated and *Cyp1a2* being significantly upregulated, and the key metabolites, linoleic acid, all cis-(6,9,12)-linolenic acid and PC (16:0/16:0), were significantly reduced. Phospholipase A2 Group IVC (*Pla2g4c*) is the coding gene of phospholipase A2 (PLA2), which is a family of enzymes that catalyze hydrolysis of phospholipid 2-Sn. PLA2 has an impact on inflammation, atherosclerosis, cardiovascular disease and liver disease [12]. Darapladib, an inhibitor of PLA2, has been used to reduce PLA2 levels in atherosclerotic lesions [13]. In this study, *Lactobacillus rhamnosus* hsryfm 1301 fermented milk showed inhibition of *Pla2g4c*, suggesting its potential as an inhibitor. *Cyp1a2* is the coding gene of cytochrome P450 protein. Cytochrome P450 protein is one of the important drug metabolism enzymes in the liver, which can participate in metabolism of various exogenous drugs and endogenous steroid hormones. In obese rats, the expression and activity of hepatic drug metabolizing enzymes, including *Cyp1a2*, were significantly reduced [14]. In the present study, *Lactobacillus rhamnosus* hsryfm 1301 fermented milk restored its expression and accelerated drug metabolism. Phosphatidylcholine (16:0/16:0) (PC (16:0/16:0)) is an important surface component of plasma lipoproteins and is closely associated with many diseases, such as metabolic syndrome and diabetes [15]. It has been shown that PC may also affect drug metabolism through enzymes (e.g., cytochrome P450 enzymes) that affect drug metabolism in vivo [16]. Linoleic acid (LA) and γ-linolenic acid (γLA) are important unsaturated fatty acids mainly involved in fatty acid metabolism and energy metabolism. High-fat diet caused fatty acid metabolism disorder and increased serum linoleic acid in rats. In this study, *Lactobacillus rhamnosus* hsryfm 1301 fermented milk promoted linoleic acid metabolism and significantly reduced serum linoleic acid levels. In the linoleic acid metabolism pathway, on the one hand, *Lactobacillus rhamnosus* hsryfm 1301 fermented milk reduced PC (16:0/16:0), and, consequently, transcription of *Pla2g4c* was downregulated and LA production was reduced; on the other hand, *Lactobacillus rhamnosus* hsryfm 1301 fermented milk upregulated transcription of *Cyp1a2* and accelerated LA metabolism, leading to a decrease in γLA and LA reduction. PC (16:0/16:0) and *Pla2g4c* were also involved in both glycerophospholipid metabolism, alpha-linolenic acid metabolism and arachidonic acid metabolism to regulate lipid metabolic homeostasis.

At the same time, fatty acid biosynthesis and fatty acid degradation were also co-enriched. The transcription of *Acsbg1* enriched in this pathway was significantly upregulated, and the metabolite oleic acid was significantly reduced. *Acsbg1* is a gene encoding long chain acyl-CoA synthetase (ACSL), which plays a key role in lipid metabolism and maintaining lipid balance. ACSL is involved in fatty acid metabolism, and both endogenous and exogenous fatty acids require activation prior to metabolism to produce the corresponding lipid acyl coenzymes [17]. Oleic acid is a monounsaturated fatty acid, and disorders of lipid metabolism can elevate serum levels of oleic acid, which, in turn, induces steatosis in a variety of cells [18]. *Lactobacillus rhamnosus* hsryfm 1301 fermented milk significantly upregulated transcription of *Acsbg1*, accelerated fatty acid metabolism, which, in turn, significantly reduced serum levels of oleic acid and restored lipid metabolism homeostasis.

In addition, this study found that glycerolipid metabolism was also regulated by *Lactobacillus rhamnosus* hsryfm 1301 fermented milk, and the transcription of *Pnpla2* was significantly upregulated and glyceric acid was significantly elevated. The enzyme encoded by the *Pnpla2* catalyzes the first step of triglyceride hydrolysis in adipose tissue and is involved in fatty acid degradation. Downregulation of the transcription of *Pnpla2* leads to a defect in the function of adipose triglyceride hydrolase, allowing abnormal deposition of triglycerides in the liver and peripheral blood granulocytes [19]. Glyceric acid, a tricarbonate formed by oxidation of glycerol, is an important metabolite in the human body, and its phosphorylation generates 3-phosphoglyceric acid, which participates in the glycolytic process [20] and also has a positive effect on regulation of liver function. *Lactobacillus rhamnosus* hsryfm 1301 fermented milk upregulated transcription of *Pnpla2* and glyceric acid content, stimulated the glycerolipid metabolism pathway, promoted degradation of fatty acids while reducing the level of triglycerides in the body and thus restored the lipid metabolic balance.

Amino acid metabolism and lipid metabolism are interacting and closely linked [21]. Glycine, serine and threonine metabolism were significantly altered in both transcriptomics and metabolomics. T*Psat1*, *Cse* and *Phgdh* transcriptions were all significantly downregulated, while L-Threonine, glyceric acid and L-Serine were all significantly increased. L-serine is a non-essential amino acid that maintains normal growth and development of cells and tissues by participating in various biosynthetic pathways in the body [22]. Glyceric acid is also an intermediate product of serine degradation [20]. L-threonine is an essential amino acid that promotes growth and development, relieves fatigue and also has good anti-fatty liver properties. The enzyme encoded by phosphoglycerate dehydrogenase (*Phgdh*) is the first key enzyme that catalyzes the serine synthesis pathway, and the enzyme encoded by phosphoserine aminotransferase 1 (*Psat1*) is the second key enzyme [22]. Phgdh is highly expressed in a variety of cancers, including obesity and liver cancer, and is closely associated with tumorigenesis and proliferation. It is a new target for tumor therapy [23]. In this study, *Lactobacillus rhamnosus* hsryfm 1301 fermented milk significantly downregulated transcription of *Psat1* and *Phgdh*, while both serum L-serine and L-threonine were significantly increased. We hypothesize that probiotics produce large amounts of serine and threonine through fermentation as well as modulation of intestinal flora, which can feed back to inhibit transcription of *Phgdh* and *Psat1* and also increase the content of glyceric acid, thus achieving stalling the onset and development of obesity. *Lactobacillus rhamnosus* hsryfm 1301 fermented milk possesses the potential to act as a *Phgdh* inhibitor, which deserves further study.

Disorders of lipid metabolism not only jeopardize the physiological functions of peripheral tissues and organs but also affect the central nervous system [24,25]. Damage to the central nervous system can cause abnormalities in hypothalamic function, which can lead to a high appetite, further causing obesity. The glutamatergic synapse pathway was significantly altered in both metabolomics and transcriptomics, with significantly downregulated *Pla2g4c* and *Prkcg* and upregulated *Gria4*, as well as significantly increased L-Glutamine and 1-Stearoyl-2-arachidonoyl-sn-glycerol.

Glutamine (Gln) is the γ-carboxy amide of glutamate and the most abundant non-essential amino acid in body fluid. In addition, glutamine is an important antioxidant that is involved in synthesis of glutathione, which has anti-oxidative stress effects and improves neurological deficits and damage [26]. Thus, probiotic intervention significantly increased serum levels of Gln and improved neurological impairment. *Prkcg* encodes a protein that is a member of the protein kinase C (PKC) family. PKC is a family of serine/threonine protein kinases consisting of multiple isozymes that are activated by Ca^2+^ and diacylglycerol in the presence of phosphatidylserine and play important roles in apoptosis, neural excitability, synaptic plasticity, cell proliferation and differentiation, gene expression and cell necrosis [27]. Release of various neuromediators and free radicals after CNS injury can lead to activation of PKC and aggravate development of CNS injury. PKC also activates PLA2, which is involved in regulation of excitatory synaptic plasticity in neurons [28]. *Lactobacillus rhamnosus* hsryfm 1301 fermented milk significantly reduced transcription of *Pla2g4c* and *Prkcg* and slowed down CNS injury. The subunit encoded by *Gria4* belongs to the AMPA family and is one of the ionotropic glutamate receptors. Internalization of *Gria4* plays an important role in fine regulation of synaptic plasticity and is closely associated with long-duration inhibition and amnesia [25]. *Lactobacillus rhamnosus* hsryfm 1301 fermented milk significantly upregulated transcription of *Gria4* and alleviated internalization of *Gria4*, which, in turn, alleviated CNS injury and modulated synaptic plasticity. Notably, 1-Stearoyl-2-arachidonoyl-sn-glycerol is an analogue of diacylglycerol (DAG), which is also an activator of PKC. However, in this study, 1-Stearoyl-2-arachidonoyl-sn-glycerol was significantly increased in serum after *Lactobacillus rhamnosus* hsryfm 1301 fermented milk intervention, but transcription of *Prkcg* was still inhibited. *Lactobacillus rhamnosus* hsryfm 1301 fermented milk showed strong *Prkcg* repression and did not repress *Prkcg* by inhibiting 1-Stearoyl-2-arachidonoyl-sn-glycerol. We found that *Lactobacillus rhamnosus* hsryfm 1301 fermented milk repressed not only the transcription of *Prkcg* but also that of *P2rx1* in the co-enriched calcium signaling pathway. The *P2rx1* gene encodes a protein belonging to the P2X family of G protein-coupled receptors, which mediate rapid and selective permeability to cations and increase intracellular Ca^2+^ concentration in the presence of ATP [29]. *Lactobacillus rhamnosus* hsryfm 1301 fermented milk may have inhibited Ca^2+^ concentration and thus *Prkcg* transcription by repressing the transcription of *P2rx1*.

Disorders of the endocrine system can also cause disorders of lipid metabolism. In particular, increased insulin secretion can stimulate increased food intake while inhibiting lipolysis, thus causing fat accumulation in the body, which, in turn, leads to disorders of lipid metabolism. PKC has been shown to be a key factor in regulation of insulin receptor function [30,31]. Hypothyroidism also causes fat accumulation and leads to disorders of lipid metabolism, and PKC also plays an important role in hypothyroidism [32]. In this study, insulin secretion, thyroid hormone synthesis and the thyroid hormone signaling pathway were also co-enriched, and it showed the effects of *Lactobacillus rhamnosus* hsryfm 1301 fermented milk on hyperinsulin secretion and hypothyroidism induced by high-fat diet. The significantly different gene on these co-enriched pathways was *Prkcg*. *Prkcg* is also involved in the immune system, signal transduction and digestive system pathways. *Lactobacillus rhamnosus* hsryfm 1301 fermented milk can alleviate lipid metabolism disorders by inhibiting *Prkcg* to regulate neural, endocrine, immune, signal transduction and digestive system pathways. *Prkcg* is an important target for *Lactobacillus rhamnosus* hsryfm 1301 fermented milk to alleviate lipid metabolism disorders.

*Lactobacillus rhamnosus* hsryfm 1301 fermented milk also showed a regulatory effect on the adipocytokine signaling pathway, while regulation of lipolysis in adipocytes was also enriched. Adipokines are able to regulate the balance between energy expenditure and intake and regulate lipid synthesis and catabolism through neuro-nuclear endocrine factors, thus maintaining the balance of lipid metabolism in the organism [33]. In particular, regarding regulation of lipolysis in the adipocytes pathway, *Lactobacillus rhamnosus* hsryfm 1301 fermented milk not only significantly upregulated transcription of *Pnpla2* to promote triglyceride hydrolysis but also significantly increased serum 1-Stearoyl-2-arachidonoyl-sn-glycerol. In vivo studies have demonstrated the effects of DAG in lowering postprandial triglycerides [34,35], increasing β-oxidation of fatty acids [36,37] and reducing body weight [38,39]. In the present study, it was also found that serum 1-Stearoyl-2-arachidonoyl-sn-glycerol was significantly elevated after *Lactobacillus rhamnosus* hsryfm 1301 fermented milk intervention, while TG and FFA were significantly decreased and rat body weight was significantly reduced. Further, 1-Stearoyl-2-arachidonoyl-sn-glycerol was also found to act as a lipid molecule with signaling functions, involved in pathways such as the immune system and signal transduction.

## 5. Conclusions

In conclusion, *Lactobacillus rhamnosus* hsryfm 1301 fermented milk can regulate the metabolic pathways of linoleic acid metabolism, glycerophospholipid metabolism, fatty acid biosynthesis, alpha-linolenic acid metabolism, fatty acid degradation, glycerolipid metabolism and arachidonic acid metabolism to restore the balance of lipid metabolism. *Lactobacillus rhamnosus* hsryfm 1301 fermented milk inhibited transcription of *Pla2g4c* to reduce fatty acid synthesis and promoted transcription of *Cyp1a2*, *Acsbg1* and *Pnpla2* to accelerate fatty acid metabolism and triglyceride hydrolysis, which, in turn, significantly reduced linoleic acid, all cis-(6,9,12)- linolenic acid, PC (16:0/16:0), oleic acid and other metabolite levels. In addition, we found that *Lactobacillus rhamnosus* hsryfm 1301 fermented milk indirectly regulates lipid metabolism through regulating amino acid metabolism, the nervous system, the endocrine system and other pathways. *Lactobacillus rhamnosus* hsryfm 1301 fermented milk inhibited transcription of *Psat1* and *Phgdh* by increasing the levels of L-Threonine, L-Serine and glyceric acid, which, in turn, regulated the amino acid metabolic pathways and alleviated onset and development of obesity. Meanwhile, *Lactobacillus rhamnosus* hsryfm 1301 fermented milk alleviated central nerve damage and modulated synaptic plasticity by increasing L-Glutamine levels and transcription of *Gria4* and repressing transcription of *Pla2g4c* and *Prkcg*. *Lactobacillus rhamnosus* hsryfm 1301 fermented milk may also inhibit Ca^2+^ concentration by suppressing transcription of *P2rx1*, which, in turn, inhibits transcription of *Prkcg*, alleviating lipid metabolism disorders by regulating immune, signal transduction and digestive system pathways while alleviating insulin hypersecretion and hypothyroidism caused by high-fat diet. *Lactobacillus rhamnosus* hsryfm 1301 fermented milk can alleviate the disorders of lipid metabolism caused by high-fat diet through multi-target synergy.

## Figures and Tables

**Figure 1 nutrients-14-04850-f001:**
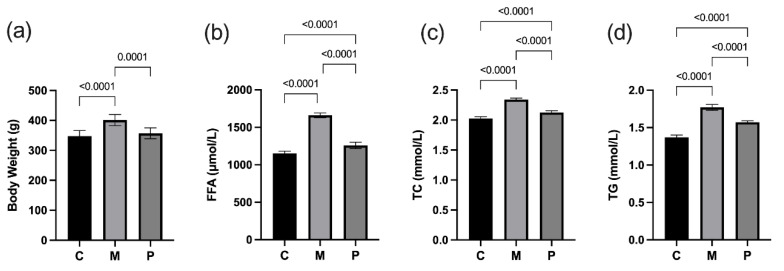
Lipid metabolism in rats. (**a**) Rats’ final body weight. (**b**) Serum FFA. (**c**) Serum TC. (**d**) Serum TG.

**Figure 2 nutrients-14-04850-f002:**
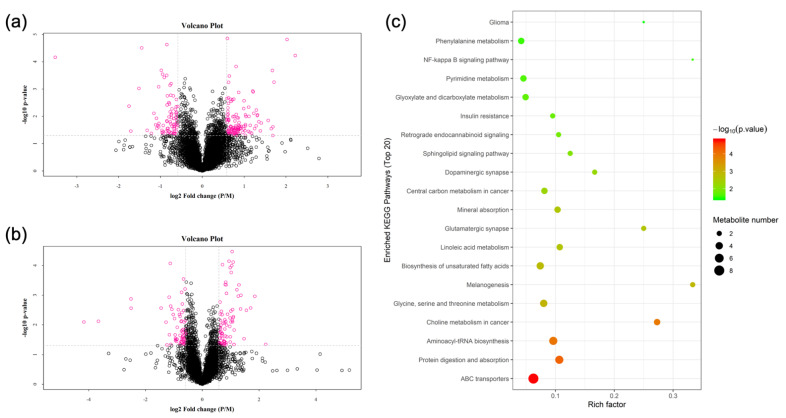
Metabolomics. (**a**) Positive ion mode differential metabolites; (**b**) negative ion mode differential metabolites; (**c**) significant KEGG pathway.

**Figure 3 nutrients-14-04850-f003:**
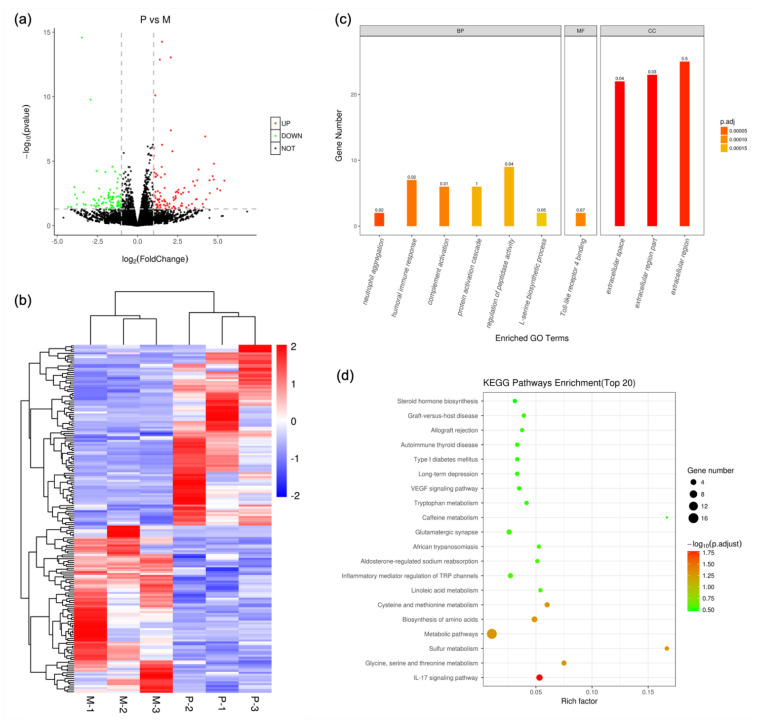
Transcriptomics. (**a**) Differential genes; (**b**) clustering analysis; (**c**) GO function annotation analysis; (**d**) significant KEGG pathway.

**Figure 4 nutrients-14-04850-f004:**
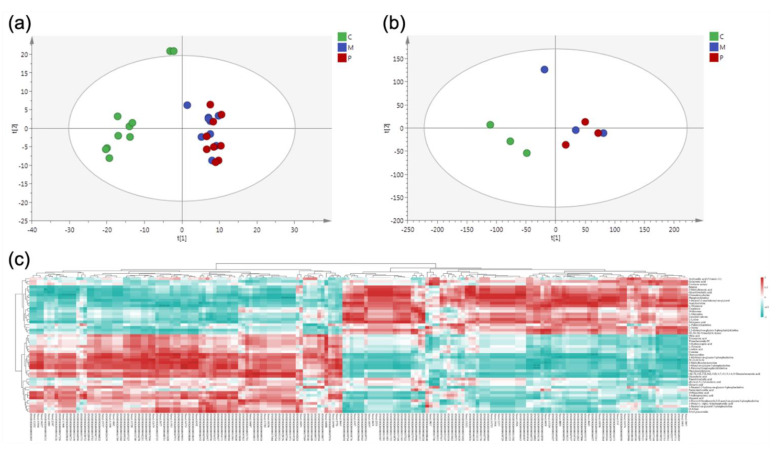
Correlation analysis. (**a**) Metabolomics PCA plot; each point represents one sample. (**b**) Transcriptomics PCA plot. (**c**) Hierarchical cluster heat map for Spearman correlation analysis of differential genes and differential metabolites. Correlation coefficient r is indicated by color. r > 0 indicates positive correlation, represented by red, r < 0 indicates negative correlation, represented by blue, and darker color indicates stronger correlation.

**Figure 5 nutrients-14-04850-f005:**
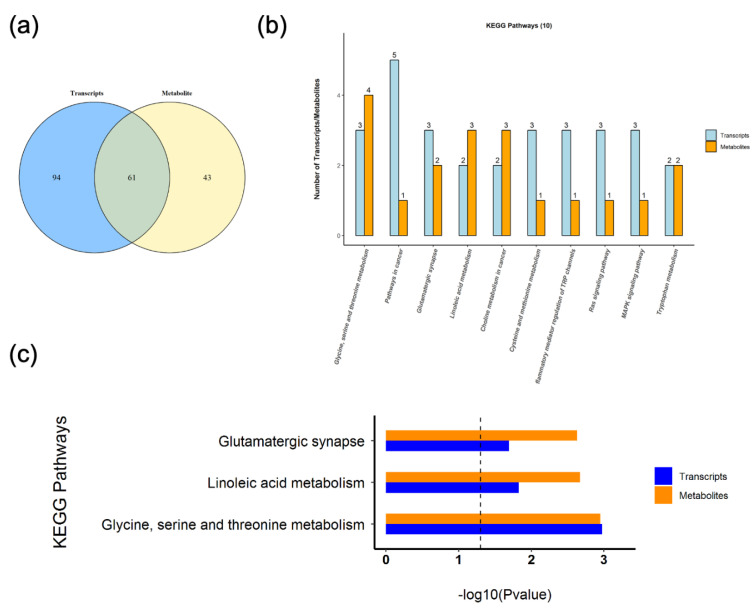
KEGG pathway joint analysis. (**a**) Venn diagram of the number of pathways involved in differential genes and differential metabolites. (**b**) KEGG pathways with the most involved genes and metabolites (top 10). (**c**) Significantly changed co-enriched KEGG pathways.

**Table 1 nutrients-14-04850-t001:** Co-enriched pathways.

Map_ID	Map Name	Gene_id	cpdName	Pathway Hierarchy
rno00591	Linoleic acid metabolism *	*Pla2g4c*, *Cyp1a2*	Linoleic acid, all cis-(6,9,12)-linolenic acid, PC(16:0/16:0)	Lipid metabolism
rno00564	Glycerophospholipid metabolism	*Pla2g4c*	PC(16:0/16:0), Phosphorylcholine,	Lipid metabolism
rno00061	Fatty acid biosynthesis	*Acsbg1*	Oleic acid	Lipid metabolism
rno00592	alpha-linolenic acid metabolism	*Pla2g4c*	PC(16:0/16:0)	Lipid metabolism
rno00071	Fatty acid degradation	*Acsbg1*	L-Palmitoylcarnitine	Lipid metabolism
rno00561	Glycerolipid metabolism	*Pnpla2*	Glyceric acid	Lipid metabolism
rno00590	Arachidonic acid metabolism	*Pla2g4c*	PC(16:0/16:0)	Lipid metabolism
rno00260	Glycine, serine and threonine metabolism *	*Psat1*, *Cse*, *Phgdh*	L-Threonine, Glyceric acid, L-Serine, Betaine	Amino acid metabolism
rno00270	Cysteine and methionine metabolism	*Psat1*, *Cse*, *Phgdh*	L-Serine	Amino acid metabolism
rno00380	Tryptophan metabolism	*Cyp1a2*, *Acmsd*	Formylanthranilic acid, Anthranilic acid (Vitamin L1)	Amino acid metabolism
rno00250	Alanine, aspartate and glutamate metabolism	*Asns*	L-Glutamine	Amino acid metabolism
rno04724	Glutamatergic synapse *	*Pla2g4c*, *Gria4*, *Prkcg*	L-Glutamine, 1-Stearoyl-2-arachidonoyl-sn-glycerol	Nervous system
rno04728	Dopaminergic synapse	*Gria4*, *Prkcg*	L-Tyrosine, 1-Stearoyl-2-arachidonoyl-sn-glycerol	Nervous system
rno04723	Retrograde endocannabinoid signaling	*Gria4*, *Prkcg*	PC(16:0/16:0), 1-Stearoyl-2-arachidonoyl-sn-glycerol	Nervous system
rno04730	Long-term depression	*Pla2g4c*, *Prkcg*	1-Stearoyl-2-arachidonoyl-sn-glycerol	Nervous system
rno04726	Serotonergic synapse	*Pla2g4c*, *Prkcg*	1-Stearoyl-2-arachidonoyl-sn-glycerol	Nervous system
rno04720	Long-term potentiation	*Prkcg*	1-Stearoyl-2-arachidonoyl-sn-glycerol	Nervous system
rno04727	GABAergic synapse	*Prkcg*	L-Glutamine	Nervous system
rno04725	Cholinergic synapse	*Prkcg*	1-Stearoyl-2-arachidonoyl-sn-glycerol	Nervous system
rno04911	Insulin secretion	*Prkcg*	1-Stearoyl-2-arachidonoyl-sn-glycerol	Endocrine system
rno04912	GnRH signaling pathway	*Pla2g4c*	1-Stearoyl-2-arachidonoyl-sn-glycerol	Endocrine system
rno04925	Aldosterone synthesis and secretion	*Prkcg*	1-Stearoyl-2-arachidonoyl-sn-glycerol	Endocrine system
rno04921	Oxytocin signaling pathway	*Pla2g4c*, *Prkcg*	1-Stearoyl-2-arachidonoyl-sn-glycerol	Endocrine system
rno04918	Thyroid hormone synthesis	*Prkcg*	1-Stearoyl-2-arachidonoyl-sn-glycerol	Endocrine system
rno04919	Thyroid hormone signaling pathway	*Prkcg*	1-Stearoyl-2-arachidonoyl-sn-glycerol	Endocrine system
rno04920	Adipocytokine signaling pathway	*Acsbg1*	1-Stearoyl-2-arachidonoyl-sn-glycerol	Endocrine system
rno04923	Regulation of lipolysis in adipocytes	*Pnpla2*	1-Stearoyl-2-arachidonoyl-sn-glycerol	Endocrine system
rno04915	Estrogen signaling pathway	*Hsp90aa1*	1-Stearoyl-2-arachidonoyl-sn-glycerol	Endocrine system
rno04916	Melanogenesis	*Prkcg*	L-Tyrosine, 1-Stearoyl-2-arachidonoyl-sn-glycerol	Endocrine system
rno04650	Natural killer cell mediated cytotoxicity	*Gzmbl2*, *Prkcg*	1-Stearoyl-2-arachidonoyl-sn-glycerol	Immune system
rno04666	Fc gamma R-mediated phagocytosis	*Pla2g4c*, *Prkcg*	1-Stearoyl-2-arachidonoyl-sn-glycerol	Immune system
rno04611	Platelet activation	*Pla2g4c*, *P2rx1*	1-Stearoyl-2-arachidonoyl-sn-glycerol	Immune system
rno04664	Fc epsilon RI signaling pathway	*Pla2g4c*	1-Stearoyl-2-arachidonoyl-sn-glycerol	Immune system
rno04062	Chemokine signaling pathway	*Ccl17*	1-Stearoyl-2-arachidonoyl-sn-glycerol	Immune system
rno04014	Ras signaling pathway	*Pla2g4c*, *Fgf21*, *Prkcg*	1-Stearoyl-2-arachidonoyl-sn-glycerol	Signal transduction
rno04010	MAPK signaling pathway	*Pla2g4c*, *Fgf21*, *Prkcg*	1-Stearoyl-2-arachidonoyl-sn-glycerol	Signal transduction
rno04071	Sphingolipid signaling pathway	*Kng1*, *Prkcg*	L-Serine, 1-Stearoyl-2-arachidonoyl-sn-glycerol	Signal transduction
rno04066	HIF-1 signaling pathway	*LOC100911372*, *Prkcg*	1-Stearoyl-2-arachidonoyl-sn-glycerol	Signal transduction
rno04370	VEGF signaling pathway	*Pla2g4c*, *Prkcg*	1-Stearoyl-2-arachidonoyl-sn-glycerol	Signal transduction
rno04015	Rap1 signaling pathway	*Fgf21*, *Prkcg*	1-Stearoyl-2-arachidonoyl-sn-glycerol	Signal transduction
rno04020	Calcium signaling pathway	*P2rx1*, *Prkcg*	1-Stearoyl-2-arachidonoyl-sn-glycerol	Signal transduction
rno04012	ErbB signaling pathway	*Prkcg*	1-Stearoyl-2-arachidonoyl-sn-glycerol	Signal transduction
rno04072	Phospholipase D signaling pathway	*Pla2g4c*	1-Stearoyl-2-arachidonoyl-sn-glycerol	Signal transduction
rno04024	cAMP signaling pathway	*Gria4*	1-Stearoyl-2-arachidonoyl-sn-glycerol	Signal transduction
rno04970	Salivary secretion	*Prkcg*	1-Stearoyl-2-arachidonoyl-sn-glycerol	Digestive system
rno04971	Gastric acid secretion	*Prkcg*	1-Stearoyl-2-arachidonoyl-sn-glycerol	Digestive system
rno04972	Pancreatic secretion	*Prkcg*	1-Stearoyl-2-arachidonoyl-sn-glycerol	Digestive system
rno05200	Pathways in cancer	*Kng1*, *Zbtb16*, *Hsp90aa1*, *Fgf21*, *Prkcg*	1-Stearoyl-2-arachidonoyl-sn-glycerol	Cancers: Overview
rno05231	Choline metabolism in cancer	*Pla2g4c*, *Prkcg*	PC(16:0/16:0), 1-Stearoyl-2-arachidonoyl-sn-glycerol, Phosphorylcholine	Cancers: Overview
rno05214	Glioma	*Prkcg*	1-Stearoyl-2-arachidonoyl-sn-glycerol	Cancers: Specific types
rno05223	Non-small cell lung cancer	*Prkcg*	1-Stearoyl-2-arachidonoyl-sn-glycerol	Cancers: Specific types
rno05143	African trypanosomiasis	*Kng1*, *Prkcg*	1-Stearoyl-2-arachidonoyl-sn-glycerol	Infectious diseases: Parasitic
rno05146	Amoebiasis	*Prkcg*	1-Stearoyl-2-arachidonoyl-sn-glycerol	Infectious diseases: Parasitic
rno04270	Vascular smooth muscle contraction	*Pla2g4c*, *Prkcg*	1-Stearoyl-2-arachidonoyl-sn-glycerol	Circulatory system
rno04540	Gap junction	*Prkcg*	1-Stearoyl-2-arachidonoyl-sn-glycerol	Cellular community
rno04750	Inflammatory mediator regulation of TRP channels	*Pla2g4c*, *Kng1*, *Prkcg*	1-Stearoyl-2-arachidonoyl-sn-glycerol	Sensory system
rno04713	Circadian entrainment	*Gria4*, *Prkcg*	1-Stearoyl-2-arachidonoyl-sn-glycerol	Environmental adaptation
rno04961	Endocrine and other factor-regulated calcium reabsorption	*Prkcg*	1-Stearoyl-2-arachidonoyl-sn-glycerol	Excretory system
rno05031	Amphetamine addiction	*Gria4*, *Prkcg*	L-Tyrosine	Substance dependence
rno00920	Sulfur metabolism	*LOC103689947*, *Impad1*	L-Serine	Energy metabolism
rno00910	Nitrogen metabolism	*Car6*	L-Glutamine	Energy metabolism

* Significantly changed KEGG pathways.

## Data Availability

The data supporting the findings reported here are available upon reasonable request from the corresponding author.

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
