# Peer review of "Effect of Lactobacillus rhamnosus hsryfm 1301 Fermented Milk on Lipid Metabolism Disorders in High-Fat-Diet Rats"

_nutrients, 2022, doi:10.3390/nu14224850_

Round 1

Reviewer 1 Report

The objective of this manuscript was to investigate the treatment mechanism of lipid metabolism disorders using fermented milk with Lactobacillus rhamnosus hsryfm 1301 by combination of transcriptomic and metabolomic analysis. The manuscript is well written and organized. The research topic is highly interesting and conncet the food biotechnology with the clinical nutrition and there is a need to use natural food products to alleviate lipid metabolism disorder. However, some major and minor changes are suggested below:

Major comments

- The objectives of the study in the introduction and abstract are not written in a clear way. Please re-write

- L53: cite some of these studies

- More literature reviews should be added to clarify the rational of your study.

- The methodology section written in a professional way, however, a separate sub-section about statistical analysis.

The results section is well written.

Discussion part should be concise and not repeating the results. Further, it should include the comparing the results with other probiotic strains.

The conclusions should be written in a separate section.

Please add recommendations of using the probiotic strain in fermentation of milk to produce dairy products. This should be added into sections of Abstract and Conclusions

Minor changes

L10-11: may help to investigate

L18: "KEGG" write in full words

L19-20 use lower-case letters in the names of fatty acids and amino acids

L44: re-number the sections start with 1. Introduction

L66: italicize Lactobacillus casei

L199: add the number of the reference "Benjamini & Hochberg"

L205: P value

L259: edit "value"

L76: glycine, serine and threonine metabolism, cystein

L425, 431, 433 etc.: hsryfm 1301

L446: thyroid

L456: adipocytokine

L460: regulation

L463: italicize "In vivo"

Author Response

Dear reviewer:

Thank you for reviewing the manuscript and giving us your valuable comments. The answers to your questions are as follows.

Major comments

Q: - The objectives of the study in the introduction and abstract are not written in a clear way. Please re-write

A: The objectives of the study in the introduction and abstract had been re-written.

Q: - L53: cite some of these studies

A: The reference had been added.

Q: - More literature reviews should be added to clarify the rational of your study.

A: The introduction had been revised to make it clear and rationally.

Q: - The methodology section written in a professional way, however, a separate sub-section about statistical analysis.

The results section is well written.

A: The statistical analysis had been written in a separate section.

Q: Discussion part should be concise and not repeating the results. Further, it should include the comparing the results with other probiotic strains.

A: The discussion had been revised.

Q: The conclusions should be written in a separate section.

A: The conclusions had been written in a separate section.

Q: Please add recommendations of using the probiotic strain in fermentation of milk to produce dairy products. This should be added into sections of Abstract and Conclusions

A: The preparation of fermented milk had already been described in the methods, so it was not added again in the abstract and conclusions.

Minor changes

Q: L10-11: may help to investigate

A: This error had been corrected.

Q: L18: "KEGG" write in full words

A: The full words had been added.

Q: L19-20 use lower-case letters in the names of fatty acids and amino acids

A: The names of fatty acids, amino acids and all the pathways in the text had been used lower-case letters

Q: L44: re-number the sections start with 1. Introduction

A: The sections start had been re-numbered.

Q: L66: italicize Lactobacillus casei

A: This error had been corrected.

Q: L199: add the number of the reference "Benjamini & Hochberg"

A: The reference had been added.

Q: L205: P value

A: This error had been corrected.

Q: L259: edit "value"

A: This error had been corrected.、

Q: L76: glycine, serine and threonine metabolism, cysteine

A: This error had been corrected.

Q: L425, 431, 433 etc.: hsryfm 1301

A: This error had been corrected.

Q: L446: thyroid

A: This error had been corrected.

Q: L456: adipocytokine

A: This error had been corrected.

Q: L460: regulation

A: This error had been corrected.

Q: L463: italicize "In vivo"

A: This error had been corrected.

Thank you and best regards.

Yours sincerely,

Hengxian Qu

Reviewer 2 Report

the manuscript is well written and presented.

Abstract is bit lengthy

Introduction: good

Material & methods: all methods have been given precisely

Result: well presented and supported with justifiable references

Conclusion: clear 

References: kindly go through the journals guidelines

Fig: clear and understandable

Minor correction are needed. Kindly go through the manuscript attached. Corrections required have been highlighted .

Author Response

Dear reviewer:

Thank you for reviewing the manuscript and giving us your valuable comments. The answers to your questions are as follows.

Q:

the manuscript is well written and presented.

Abstract is bit lengthy

Introduction: good

Material & methods: all methods have been given precisely

Result: well presented and supported with justifiable references

Conclusion: clear

References: kindly go through the journals guidelines

Fig: clear and understandable

Minor correction are needed. Kindly go through the manuscript attached. Corrections required have been highlighted

A:

The abstract had been revised and shortened. The article had been subjected to a minor correction.

Thank you and best regards.

Yours sincerely,

Hengxian Qu